# Semi-Supervised Anomaly Detection of Dissolved Oxygen Sensor in Wastewater Treatment Plants

**DOI:** 10.3390/s23198022

**Published:** 2023-09-22

**Authors:** Liliana Maria Ghinea, Mihaela Miron, Marian Barbu

**Affiliations:** 1Department of Automatic Control and Electrical Engineering, Faculty of Automation, Computers, Electrical Engineering and Electronics, Dunărea de Jos University of Galați, 47 Domnească Str., 800008 Galați, Romania; liliana.ghinea@ugal.ro (L.M.G.); marian.barbu@ugal.ro (M.B.); 2Department of Computer Science and Information Technology, Faculty of Automation, Computers, Electrical Engineering and Electronics, Dunărea de Jos University of Galați, 47 Domnească Str., 800008 Galați, Romania

**Keywords:** wastewater treatment process (WWTP), semi-supervised learning (SSL), autoencoder (AE), Isolation Forest (IF), Local Outlier Factor (LOF), One-Class Support Vector Machine (OCSVM)

## Abstract

As the world progresses toward a digitally connected and sustainable future, the integration of semi-supervised anomaly detection in wastewater treatment processes (WWTPs) promises to become an essential tool in preserving water resources and assuring the continuous effectiveness of plants. When these complex and dynamic systems are coupled with limited historical anomaly data or complex anomalies, it is crucial to have powerful tools capable of detecting subtle deviations from normal behavior to enable the early detection of equipment malfunctions. To address this challenge, in this study, we analyzed five semi-supervised machine learning techniques (SSLs) such as Isolation Forest (IF), Local Outlier Factor (LOF), One-Class Support Vector Machine (OCSVM), Multilayer Perceptron Autoencoder (MLP-AE), and Convolutional Autoencoder (Conv-AE) for detecting different anomalies (complete, concurrent, and complex) of the Dissolved Oxygen (DO) sensor and aeration valve in the WWTP. The best results are obtained in the case of Conv-AE algorithm, with an accuracy of 98.36 for complete faults, 97.81% for concurrent faults, and 98.64% for complex faults (a combination of incipient and concurrent faults). Additionally, we developed an anomaly detection system for the most effective semi-supervised technique, which can provide the detection of delay time and generate a fault alarm for each considered anomaly.

## 1. Introduction

Anomaly detection is a popular machine learning technique used to identify unusual or rare instances in data. It can be used in many different fields where detecting abnormalities or outliers is essential for maintaining the system’s health, sustaining security, or optimizing processes. There are various examples of problems involving fault detection, such as frauds [1], network intrusions [2], manufacturing defects [3], anomaly detection in time series data [4], cybersecurity [5,6], microfluidics [7,8,9,10], and most importantly, anomaly detection in wastewater treatment plants [11,12,13].

Wastewater treatment plants play a vital role in modern society due to their significant importance in addressing environmental and public health challenges. The purpose of these facilities is to eliminate pollutants and contaminants from wastewater before it is discharged into the environment. As a result, strict regulations and permissions outline the acceptable discharge limits of pollutants into water bodies.

Unfortunately, the complex and dynamic processes from WWTPs [14] have numerous variables which can create an environment where different types of failures (e.g., complete, concurrent, complex) can occur. Such failures (e.g., mechanical, biological) can disrupt the treatment process, potentially leading to environmental harm, regulatory non-compliance, and compromised public health. Hence, the demand for advanced anomaly detection systems in WWTPs has become very important in recent years. Through these, the operators can maintain the effective operation of WWTPs by taking prompt corrective actions, preventing disruptions to the treatment process, minimizing potential environmental impacts, and ensuring that the treated effluent meets the regulatory standards.

The application of artificial intelligence (AI) techniques for detecting faults has significantly advanced. This technological approach has provided a substantial boost to fault detection capabilities, particularly in complex systems like WWTPs. From the AI-driven fault detection techniques, we have analyzed in this study five semi-supervised learning algorithms: Isolation Forest (IF), Local Outlier Factor (LOF), One-Class Support Vector Machine (OCSVM), Multilayer Perceptron Autoencoder (MLP-AE) and Convolutional Autoencoder (Conv-AE). The most complex are autoencoders (AEs), which operate as neural networks. They are used to learn and represent the normal behavior of the system based on historical data. Subsequently, they identify deviations from this learned norm, effectively detecting anomalies [15].

The AE architecture consists of two main parts: the encoder and the decoder. The encoder compresses the input data into a lower-dimensional representation, often referred to as a bottleneck or latent space. The decoder then attempts to reconstruct the original input from this compressed representation [16]. Once the AE is trained, it can be used to process new, unseen data. The reconstruction loss between the input and its reconstruction is calculated for each data point. A threshold is set to differentiate between normal and anomalous instances. Anomalies are identified when data points with reconstruction losses exceed the threshold.

AE’s ability to learn complex patterns and detect subtle deviations makes them well-suited for anomaly detection in WWTPs. They do not require the explicit labeling of all data, allowing them to adapt to changing conditions and new types of anomalies. However, successful implementation requires the careful consideration of model architecture, hyperparameters, training data quality, and threshold selection to achieve accurate and effective anomaly detection within WWTP operations. But even though there are remarkable results from semi-supervised anomaly detection in other sectors, the application of autoencoders in the context of WWTPs remains relatively unexplored. The scarcity of studies underscores the need for further research and exploration in this domain.

Therefore, in our study, we analyzed five semi-supervised methods for anomaly detection of the Dissolved Oxygen (DO) sensor and aeration valve from WWTPs. For the best semi-supervised technique, a fault detection system was proposed. The main contribution lies in implementing the semi-supervised fault detection system that leverages the maximum Mean Absolute Error (MAE) loss value from the training data to set a dynamic threshold for anomaly detection. Unlike traditional static thresholds, which may not adapt well to changing data patterns, this method employs a data-driven approach to establish a threshold that reflects the current data distribution. The uniqueness of our study lies in the capability of our anomaly detection system to identify not only complete faults, as commonly presented in the literature, but also concurrent and complex faults in WWTPs. Moreover, the proposed fault detection system introduces another key advantage by incorporating detection delay times and generating specific fault alarms for each type of anomaly. This means that our system not only detects issues at various stages of development, but also provides timely alerts, allowing for proactive intervention. This proactive aspect ensures that potential faults are addressed in a timely manner, minimizing their impact, and contributing to a more robust and responsive fault detection system.

The paper is organized as follows: Section 2 presents the related works. Section 3 refers to the materials and methods used, such as the proposed framework, major contributions of our work, descriptions of the considered anomaly scenarios, data preprocessing from datasets, statistical analysis of training data, five semi-supervised algorithms, hyperparameter tuning, threshold procedure selection, and evaluation criteria of the proposed algorithms. Section 4 discusses and reveals the results of our research. Section 5 is dedicated to the conclusions.

## 2. Related Work

There are many works that deal with the anomaly detection of sensors in WWTPs. From the fault detection techniques, the most widely explored are: (1) statistical techniques, such as Principal Component Analysis (PCA) and Partial Least Squares (PLS), and (2) machine learning algorithms (MLs), such as Gaussian Process Regression (GPR) and Artificial Neural Networks (ANNs) (e.g., Long Short-Term Memory (LSTM) networks, autoencoders (AEs), and Radial Basis Function (RBF) neural network).

From the first category, PCA and PLS were widely applied for anomaly detection in WWTPs. For example, the authors of [17] developed a real-time fault detection and isolation (FDI) system by simulating the BSM1. Also, in [10,18], these statistical methods were used to detect and monitor failures in WWTPs.

From the second category, GPR is a non-parametric Bayesian approach to regression, which has been used many times throughout papers to identify faults and anomalies in WWTPs. For instance, in paper [19], GPR was used to monitor the Biochemical Oxygen Demand (BOD) value of the WWTP effluent. The BOD value was used to predict the Sludge Volume Index (SVI) and the occurrence of filamentous sludge bulking. This method identified the faults studied in the paper with an accuracy of 95.5%. Another notable work is paper [20], that proposes two GPR models: one is GPR with maximum likelihood estimation (GPR-MLE) and the other is GPR using Monte Carlo sequential estimation (GPRSMC). The methods both evaluate the estimation of missing values in the WWTP flow rate and find the most efficient method for detecting drift faults in the values of sensors that measure ammonia levels. The experiments were conducted on a phenomenological influent simulator by using real data and the result obtained showed a 74.5% drift detection.

With regard to ANNs, paper [21] presents a Feedforward Neural Network (FNN) algorithm for isolating multiple anomalies (six types of failures such as recirculation pump, supply pump, excess of the sludge pump, biomass concentration sensor, DO concentration sensor, and partial (25%) supply pump) in WWTPs. The method proved to be efficient as it recognized the anomalies with a 97.2% accuracy. Another notable paper is [22], where LSTM networks are used to identify collective failures in the sensors of a WWTP. The data are collected and labeled as normal and faulty. The results obtained with LSTM networks are compared with other methods, such as PCA and SVM models, and show that the anomalies can be identified with a 92% accuracy. The research of [23] addresses the need for efficient wastewater treatment plant (WWTP) operation by monitoring influent conditions (ICs) to detect potential anomalies. It introduces kernel machine learning models, specifically the kernel principal components analysis-based One-Class Support Vector Machine (KPCA-OCSVM), to classify ICs and identify anomalies in a seven-year multivariate IC time series. These kernel-based algorithms outperform previous linear PCA-based models, offering improved anomaly detection capabilities while maintaining computational efficiency and making them adaptable for different WWTPs. Additionally, others [24] emphasize the significance of monitoring the operation cost index (OCI) in WWTPs for financial planning and operational optimization. They introduce four predictive models, including ARMA variants with recursive least squares (RLS) and recursive extended least squares (RELS), as well as nonlinear auto-regressive neural networks (NARNN) and nonlinear auto-regressive neural networks with external input (NARXNN). Among these models, the nonlinear NARXNN demonstrates superior predictive capabilities, particularly in handling the inherent nonlinearity of wastewater treatment processes. The studies [25,26] apply an RBF neural network in order to detect faults of the DO sensor in Benchmark Simulation Model No. 1 (BSM1). Other studies [27,28,29] proposed autoencoders for the fault detection of failures like abrupt changes or drift sensors by using BSM1. In [30], a variational autoencoder (VAE) is applied to address fault detection, such as sludge expansion fault and small-magnitude variable step, by taking into consideration the dynamic changes within the treatment process. Furthermore, Ref. [30] proposed two autoencoders, Convolutional (Conv) and Long Short-Term Memory (LSTM) autoencoders, to identify failures (drift, bias, precision degradation, spike, and stuck) of the DO sensor in WWTPs. There were used in three different scenarios by varying the occurrence order, intensity, and duration of faults. The metrics demonstrated that Conv-AE has a better performance than LSTM-AE.

Overall, it becomes obvious that machine and deep learning methods, especially autoencoders and convolutional neural networks (CNNs), have gained significant importance in various engineering applications, particularly in fault and damage detection. For example, a recent study [31] proposed to combine deep stacked autoencoders (SAEs) with multi-sensor fusion to enhance the accuracy of damage diagnosis in concrete structures. Another [32] presents an efficient one-dimensional convolutional gated recurrent unit neural network (1D-CGRU) for real-time structural damage detection, combining 1D-CNN for spatial feature extraction and GRU for temporal mapping. All these demonstrate that there is enough place for autoencoders and CNNs to continue to advance the field of engineering by providing more accurate, efficient, and versatile tools for fault and damage detection.

## 3. Materials and Methods

This paper is focused on detecting the faults of the DO sensor (bias, drift, spike, and precision degradation (PD)) and aeration valve from WWTPs. We selected these faults because DO sensors and aeration valves are key components in these complex processes. For instance, DO sensors monitor the amount of oxygen dissolved in the wastewater, an important parameter for biological treatment processes. On the other hand, aeration valves control the supply of oxygen to support microbial activity. Thus, early detection of faults in these components becomes vital for maintaining process efficiency, resource optimization, regulatory compliance, and equipment protection.

In this research, we conducted several experiments in order to collect data with different fault scenarios (complete, concurrent, and complex anomalies) by using the Benchmark Simulation Model No. 2 (BSM2), developed by the IWA Task Group [33]. Each scenario corresponds to a test dataset shown in Table 1. In total, our study was conducted using three test datasets: (1) complete faults; (2) concurrent faults; and (3) complex faults (a combination of incipient and concurrent faults). Two processing steps were then applied to the data from the datasets: (1) data cleaning and (2) normalization by feature extraction (e.g., mean and standard deviation for generating training and testing sequences). Then, each dataset was used to feed five semi-supervised learning techniques for anomaly detection such as Isolation Forest (IF), Local Outlier Factor (LOF), One-Class Support Vector Machine (OCSVM), Multilayer Perceptron Autoencoder (MLP-AE), and Convolutional Autoencoder (Conv-AE).

As shown in Section 4, the performance of the semi-supervised methods was computed on each dataset and the best result was obtained for Conv-AE. Also, a fault detection system was designed for the best semi-supervised method. This was performed by determining the threshold using a procedure that finds the maximum MAE loss value on the training data. If the reconstruction loss obtained from the testing data is greater than the threshold from the previous step, then an anomaly is detected. The detection time and the alarm signal were obtained for each anomaly from the datasets involved in this study.

### 3.1. Proposed Framework

The main objective of our research is to build an advanced semi-supervised anomaly detection system for DO sensor and aeration valve, which are key components within WWTPs. In Figure 1, the block diagram of the proposed framework for anomaly detection in WWTPs is presented. This represents a concise conceptual block diagram outlining the key steps of our research, as follows:Collect ‘normal’ and ‘anomalous’ data representing different fault scenarios (complete, concurrent, and complex) for DO sensor and aeration valve. In total, three test datasets are prepared.Preprocess the data from the datasets (data cleaning and normalization) and generate training and testing sequences.For each scenario, test five semi-supervised machine learning techniques for anomaly detection: Isolation Forest, Local Outlier Factor, One-Class Support Vector Machine, MLP Autoencoder, and Convolutional Autoencoder.For each algorithm, compute the evaluation metrics as accuracy, precision, recall, and F1-score.Develop a fault detection system with the best semi-supervised method where: The threshold is determined by a procedure that finds the max MAE loss value on the training data. If the reconstruction loss obtained on the testing data is greater than the threshold from the previous step, then an anomaly is detected. In other words, if the data have an MAE loss greater than the threshold, they are likely to deviate significantly from the norm and become potential anomalies.Detection time delays and anomaly alarms are generated for each type of anomaly.

### 3.2. Data Description

Raw data of both the normal and anomalous state of the DO sensor and aeration valve are collected in three datasets (presented in Figure 2). Each dataset represents a different anomaly scenario (complete, concurrent, and complex) of the DO sensor and aeration valve.

In our experiments (Figure 2), we analyzed different types of anomalies in the DO sensor and aeration valve. For each, an anomaly window was selected:

One anomaly window of the aeration valve, consisting of 20 days. This is directly linked to Oxygen Transfer Coefficient ($K_{L}a_{4}$) of the bioreactor aeration system. When anomalies are affecting the aeration valve, it indirectly produces erroneous DO sensor outputs.One anomaly window for DO sensor drift of 30 days. This can be observed as a gradual deviation, leading to a shift in the sensor’s output values.One anomaly window for DO sensor bias of 25 days. In our study, this is characterized by a constant difference between the true value and the faulty DO sensor output of +1.5 mg/L.Four anomaly windows for DO sensor spike consisting of 2 days. Spike anomalies are large amplitude peaks (e.g., 2, 2.5, 1.5, and 2.7) at constant time intervals.One window for a DO sensor precision degradation of 30 days. Precision degradation in a DO sensor emerges as a loss in the precision of the sensors or control systems used for supervising and managing the treatment procedure.

The anomaly windows were combined to result in test dataset 1, test dataset 2, and test dataset 3. Table 1 presents the datasets that were considered and the starting day and the fault duration for each anomaly.

The first scenario (test dataset 1) contains complete anomalies of the aeration valve and DO sensor (drift, bias, spike, and precision degradation (PD)). The second scenario considers concurrent anomalies, a simultaneous occurrence of the following cases: aeration valve fault and sensor drift fault; aeration valve fault and sensor bias fault; aeration valve and sensor spike fault; and aeration valve fault and sensor PD. And the third scenario refers to complex anomalies (a combination of concurrent and incipient faults) between the DO sensor and the aeration valve. These types of faults could lead to many operational challenges and negative impacts on the treatment process. More precisely, this dataset is formed by the following combinations: 80% aeration valve fault + 20% sensor drift fault; 80% aeration valve fault + 20% sensor bias fault; 80% aeration valve fault + 20% sensor spike fault; and 80% aeration valve fault + 20% sensor PD fault. 

### 3.3. Data Preprocessing and Splitting

To enhance the accuracy of the semi-supervised learning algorithms, the data were preprocessed with the following procedures: data cleaning and normalization. Data cleaning is used to identify and rectify inconsistencies, inaccuracies, and missing values in the datasets to ensure the data’s quality and reliability before analysis. The normalization is performed with the Z-score normalization [34], as follows:(1)$$\mu =\frac{x_{1}+x_{2}+\ldots +x_{n}}{n}$$
where μ is the mean of a dataset with *n* values ($x_{1},x_{2},\ldots ,x_{n})$.
(2)$$\sigma =\sqrt{\frac{{\left(x_{1}-\mu \right)}^{2}+{\left(x_{2}-\mu \right)}^{2}+\ldots +{\left(x_{n}-\mu \right)}^{2}}{n}}$$
where σ is the standard deviation of a dataset with *n* values ($x_{1},x_{2},\ldots ,x_{n})$.

After preprocessing, the datasets are split into training and testing with a 0.8:0.2 ratio for classical semi-supervised algorithms. In the case of autoencoders, separate datasets for training and testing are prepared. All semi-supervised methods use the training dataset with only normal instances, and the testing datasets with both normal and anomalous instances. Some of the data are labeled (with normal and anomaly classes). The performance evaluation is conducted using metrics like the confusion matrix, accuracy, precision, recall, and F1-score. The selected algorithms are configured to operate in a semi-supervised manner [35], requiring a careful balance between the labeled and unlabeled data.

### 3.4. Statistical Analysis of Training Data

In this study, we conducted a comprehensive statistical analysis on the training dataset using the Pandas Python library shown in Table 2. This library played a central role because it allowed us to efficiently explore, clean, and prepare the dataset for subsequent tasks. Using the *describe()* function, we extracted the statistical data presented in Table 2. This information is valuable for understanding the distribution and characteristics of the data, especially in the initial stages of data exploration and analysis.

Each statistics provided specific information, as follows: “count” indicates the number of non-null values in the dataset; “mean” offers a measure of central tendency; “standard deviation” (std) indicates the dispersion of data around the mean; “minimum” (min) represents the lowest data point; and the “maximum” (max) finds the highest data point. These statistics collectively helped us to understand the distribution and characteristics of the data.

### 3.5. Classical Semi-Supervised Methods

Isolation Forest (IF) is an anomaly detection method that efficiently identifies outliers in datasets, and was first introduced in [36]. The algorithm uses a recursive binary tree structure to isolate anomalies efficiently. The anomaly score for each data point is based on how quickly it can be isolated in these trees. The formula to calculate the anomaly score for a data point X in the Isolation Forest [34] is as follows:(3)$$s\left(X,\;n\right)\;={\;2}^{-\frac{E(h\left(X\right))}{c(n)}}$$
where s(X, n) is the anomaly score for data point X in the Isolation Forest with n trees; $E\left(h\left(X\right)\right)$ is the average path length for data point X across all isolation trees; c(n) is a constant term used to normalize the score and is calculated as $c\left(n\right)=2$·${log}_{2}\left(n-1\right)-(2\frac{n-1}{n})$, where n is the number of data points used to build the tree.

The anomaly score s(X, n) measures how quickly data point X was isolated in the tree compared to the average isolation path length of all points in the tree. Lower scores indicate that a data point was isolated quickly, suggesting it is likely an anomaly, while higher scores indicate that it required more steps to isolate, suggesting it is more likely a normal data point. This approach is particularly useful for high-dimensional data and is capable of handling both global and local anomalies, offering a fast and effective solution for anomaly detection tasks. Therefore, this approach is computationally efficient because it only needs to build a small number of trees to produce accurate anomaly scores. Moreover, it does not require considerable hyperparameter tuning, thus using and deploying it is rather simple. However, the algorithm’s performance can change based on the features of the dataset and the parameter selection [37].

Another semi-supervised method that was applied in this research is Local Outlier Factor (LOF). This approach is used for outlier detection in data analysis and machine learning and was first introduced in paper [14]. LOF is a measure of the degree to which a data point is an outlier within its local neighborhood compared to the density of its neighboring data points. The formula for calculating the LOF of a data point X is as follows:(4)$$LOF\left(X\right)=\frac{Density(X)}{Average\;Density\;in\;Neighborhood(X)}$$
where $LOF\left(X\right)$ is the Local Outlier Factor for data point X; $Density\left(X\right)$ represents the density of data point X, which is often calculated as the inverse of the average distance between X and its k-nearest neighbors; and Average Density in Neighborhood(X) is the average density of the k-nearest neighbors of X.

LOF compares the density of a data point to the average density of its neighbors. An LOF significantly greater than 1 indicates that the point is an outlier, as its density is much lower than that of its neighbors, while an LOF close to 1 suggests that the point is similar in density to its neighbors and is not an outlier. LOF is effective at finding anomalies that might not stand out in the overall dataset but appear strange in their local surroundings. This makes LOF particularly useful when anomalies are scattered throughout the data rather than being all in one place. The advantage is that the outliers can be effectively identified in various types of datasets, even in the presence of noise or complex data structures [38]. It is a versatile and robust technique for semi-supervised outlier detection; however, it is also computationally expensive, and its results may lack interpretability, which can make it challenging to understand the reasons behind the outlier detections.

The third algorithm chosen for this research is One-Class Support Vector Machine (OCSVM), a semi-supervised machine learning method used for outlier detection and novelty detection tasks. This algorithm was first proposed in [39], where the authors extended the idea of SVMs from the standard binary classification problem to the One-Class problem, focusing on identifying anomalies or outliers in a dataset when only data from the normal type class are available for training. Learning a decision boundary that includes the majority of the typical data points in a high-dimensional feature space is the main objective of OCSVM. The formula for OCSVM [37] can be represented as follows:(5)$$\mathrm{Minimize}:\;\frac{1}{2}{\left||w|\right|}^{2}-\rho ,\;\mathrm{subject}\;\mathrm{to}\;w^{T}\cdot \phi (x_{i})\geq \;\rho$$
where w is the weight vector; $\phi (x_{i}$) represents the feature mapping of the data point $x_{i}$; and ρ is a parameter representing the offset of the decision boundary.

In simpler terms, OCSVM seeks to minimize the complexity of the decision boundary (represented by $\frac{1}{2}{\left||w|\right|}^{2}$) while ensuring that all normal data points ($x_{i}$) are on the correct side of the boundary, where $w^{T}\cdot \phi (x_{i}$) is greater than or equal to ρ. Data points that fall on the correct side of the boundary are considered normal, while those on the opposite side may be potential outliers. It operates under the assumption that typical data points are centered in a particular area, whereas outliers are dispersed over the rest of the data. OCSVM can accurately capture the distribution of the normal class by locating a hyperplane that divides the normal data points from the origin or center of the feature space [40]. The method’s ability to handle nonlinear data using kernel functions makes it a versatile and effective tool for various outlier detection applications; however, OCSVM assumes that the majority of the data points are from the normal class and that outliers are relatively scarce. In cases where the data are highly imbalanced, and the number of outliers is comparable to or even greater than the number of normal data points, OCSVM may struggle to identify the outliers accurately.

### 3.6. Semi-Supervised Deep Learning Methods

Apart from the classical semi-supervised methods, this paper also analyzes two deep learning approaches. One of them is the Multilayer Perceptron Autoencoder (MLP-AE), a powerful neural network architecture used for semi-supervised learning tasks, such as anomaly detection, that incorporates multiple hidden layers in both the encoder and decoder. The training process of the MLP-AE involves introducing the input data to the network, passing it through the encoder and decoder, and comparing the reconstructed output with the original input. Thus, the architecture of an MLP-AE consists of an input layer containing the input data, multiple hidden layers that comprise the encoder, bottleneck, and decoder layers—where the dimensionality reduction gradually takes place—and an output layer, which is expected to be as close to the input as possible [41].

The other deep learning method used for semi-supervised anomaly detection in this work is Convolution Autoencoder (Conv-AE), an approach where the architecture aims to learn a compressed representation of the input data by reducing its dimensionality and then reconstructing the original data from this compressed representation. The Conv-AE consists of two main parts: (1) the encoder, which takes the input data and compresses them, along with the final encoder layer, the bottleneck layer, which represents the compressed representation of the input data; and (2) the decoder, which takes the compressed representation (output of the bottleneck layer) and gradually upsamples it through a series of upsampling and convolutional layers, effectively reconstructing the original input data. The architecture of the decoder is frequently the exact opposite of that of the encoder, enabling the network to learn a useful representation of the data that can be used to reconstruct the input. Reconstruction loss, a metric that expresses the disparity between input and output data, is minimized during training. Depending on the type of data, binary cross-entropy (BCE), mean squared error (MSE), and Mean Absolute Error (MAE) are three common loss functions used for this purpose [42].

The architectures of the autoencoders we used in our study are presented in Figure 3.

The hyperparameters used in the SSLs are presented in Table 3 and Table 4. The process of selecting these values was facilitated through Python tuning libraries such as Keras Tuner and GridSearchCV, which ultimately led to finding the values that yielded optimal performance for each SSL algorithm. In this way, a fair comparison and analysis were performed between the considered SSLs.

We developed the algorithms for all five semi-supervised learning methods in Google Colaboratory (Colab) environment with the Python open-source libraries: Scikit-Learn 1.2.2 and TensorFlow 2.12 with Keras, a high-level deep learning API-integrated library. The advantage is that Colab offers accessibility, collaboration, and pre-installed libraries, eliminating setup complexities. With integrated GPUs, Colab facilitates efficient analysis, code sharing, and documentation, streamlining the implementation process.

### 3.7. Anomaly Detection System

From the five SSLs, the algorithm with the best evaluation metrics is selected and an advanced anomaly detection system is designed. This generates the alarm and determines the delay detection time (the time delay between the moment when the anomaly occurs and the moment when the anomaly is identified by SSL algorithm). In our study, the anomaly detection system is tested on all three anomaly scenarios but can also be successfully applied to real-time data streams if a sliding window or time interval is defined for processing the input data [30,43,44]. Furthermore, as stated in [44], real-time fault detection and diagnosis (FDD) hold significant practical importance, but the exploration of time delay remains relatively limited in prior research.

In our study, the anomaly detection system follows the below steps:

Step 1: Generate two ‘true’ binary signals (1—normal and 0—anomaly) associated to test data and predict binary signal for the SSL anomaly detection.

Step 2: Activate the alarm when the threshold is exceeded and compute the time delay by comparing the ‘true’ binary signal with the ‘predicted’ signal.

The alarm signal is:(6)$$Alarm=\left\{\begin{array}{c} 1\;if\;data>threshold\\ 0\;otherwise\; \end{array}\right.$$

The time delay between the occurrence of an anomaly and its identification by the SSL algorithm can be expressed as:(7)T_delay=T_anomaly_SSL−T_anomaly_dataset 
where T_delay is the duration between anomaly occurrence and anomaly detection by SSL; T_anomaly_SSL is the moment when the SSL algorithm identifies the anomaly; and T_anomaly_dataset is the moment when the anomaly actually occurs.

## 4. Results and Discussion

In this section, we present the performance metrics of the five semi-supervised techniques (IF, LOF, OCSVM, MLP-AE, and Conv-AE) for anomaly detection in the scenarios (complete, concurrent, and complex anomaly) described in Section 3.2. For the evaluation step (Table 5 and Table 6) we used Python documentation [45,46] to compute accuracy, precision, recall, F1-score metrics, and confusion matrices [47].

The confusion matrices from Table 6 show that classical semi-supervised anomaly detection algorithms like Isolation Forest (IF), Local Outlier Factor (LOF), and One-Class SVM are not performing as well as autoencoders when it comes to unbalanced datasets, especially when anomalies are a very small fraction of the total data.

In all scenarios (Table 5 and Table 6), Conv-AE proved to be the most efficient semi-supervised technique by achieving: (1) test dataset 1 (complete scenario) with 98.36% accuracy, 98.42% precision, 98.36% recall, and 98.38% F1-score; (2) test dataset 2 (concurrent scenario) with 97.81% accuracy, 97.80% precision, 97.81% recall, and 97.80% F1-score; and (3) test dataset 3 (complex—a combination of incipient and concurrent scenario) with 98.64% accuracy, 98.66% precision, 98.64% recall, and 98.65% F1-score. The worst results are in the case of the LOF for each scenario: (1) test dataset 1 with 73.09% accuracy, 74.97% precision, 73.09% recall, and 73.98% F1-score; (2) test dataset 2 with 75.36 accuracy, 76.97% precision, 75.36% recall, and 76.14% F1-score; and (3) test dataset 3 with 75.29% accuracy, 76.91% precision, 75.29% recall, and 76.07% F1-score. By comparing the results obtained from two autoencoders and those from the classical semi-supervised methods, AEs offer significantly enhanced efficiency in addressing anomaly detection challenges within WWTPs.

Next, the alarms are generated and the detection time for each anomaly is computed with a detection system designed to use the best semi-supervised algorithm, the Conv-AE. This system is tested in the case of all three test datasets (Figure 4, Figure 5 and Figure 6).

The threshold is determined based on the maximum MAE loss value obtained after training the Conv-AE. The threshold will determine the point beyond which a sample is classified as an anomaly. The formula for calculating the MAE loss for each sample in the test dataset using a reconstruction comparison between the predicted and actual data points is:(8)$$train\text{\_}MAE\text{\_}loss=\frac{1}{n}\sum_{i=1}^{n}\left|x_{train\text{\_}pred}^{\left(i\right)}-x_{test}^{\left(i\right)}\right|$$
where n is the number of samples in the training dataset, $x_{train\text{\_}pred}^{\left(i\right)}$ represents the predicted data from sample i in the training dataset, and $x_{train}^{i}$ represents the actual data for the same sample.

And the threshold is obtained with:(9)threshold=max⁡(train_MAE_loss)
where train_MAE_loss is the array of Mean Absolute Error (MAE) loss values calculated for each sample in the training dataset. In our study, the value of the threshold is set to 0.2588 (test dataset 1), 0.3948 (test dataset 2), and 0.4865 (test dataset 3).

The alarm and delay time are obtained as described in Section 3.7. When the Conv-AE binary signal from Figure 4, Figure 5 and Figure 6 (highlighted with orange color) becomes 0, then a fault is detected and the alarm is activated (1—active, 0—inactive). The primary task is to capture the first anomalous instance from an anomaly window. Due to data variability and subtle fluctuations, the identification of anomalies like sensor Precision Degradation (PD) can occasionally pose challenges. As illustrated in Figure 4, Figure 5 and Figure 6, this can cause the Conv-AE to occasionally overlook instances within a specific window. To address this challenge, our detection system focuses only when Conv-AE binary signal is first transitioning from value 1 to 0 during an anomaly window.

The time is computed for each anomaly with the Conv-AE detection system. This procedure holds significant importance due to its ability to provide insights into the efficiency, effectiveness, and real-world impact of the anomaly detection system. It assesses the system’s responsiveness to anomalies, thereby mitigating potential damage, enhancing system reliability, and improving overall operational efficiency.

In Table 7, the time delay of each anomaly from the three scenarios, generated by the Conv-AE detection system, is shown. The time delays obtained are generally good in the case of all scenarios (complete, concurrent, complex anomalies). From all types of anomalies, spike faults are the easiest to detect as the time delay is below 1 h. As the anomaly scenario becomes more intricate, the time delay experiences variations but not significantly. The worst time delays are 11 h in the case of 80% aeration valve + 20% sensor drift (test dataset 3—complex faults) and 6 h in the case of aeration valve + sensor drift (test dataset 2—concurrent fault).

Overall, the time delays prove that the anomaly detection system that we designed is efficient and can detect outliers in a timely manner, thus ensuring the well-function of the WWTP. Also, comparing our results with other works [26,30,46] demonstrated that our anomaly detection system is well-planned and effective. For example, in [30], a time delay of 4.41 h for drift fault is obtained, while our system is able to detect the same fault with a delay of only 3.84 h; for the bias fault, their delay is 2.5 h, while ours is significantly smaller at 0.72 h. Moreover, with the Conv-AE, we achieve higher accuracy than other studies.

The running time of the fault detection system is another critical factor in its practical application. In our evaluation, we measured the total time taken for anomaly detection in different scenarios. In the first scenario, the system demonstrated remarkable efficiency, completing the detection process in only 0.000029 min. The second scenario, while slightly longer at 0.000052 min, still maintained a swift response time. In the third scenario, the detection system performed admirably, with a runtime of 0.000047 min. These results demonstrate that the detection system is performant and can boost the early detection of anomalies in WWTPs. However, this study places its attention on a limited set of fault types covered by three datasets. In the future, it would be advantageous to explore a more diverse range of mechanical or even biological faults to evaluate how effectively the fault detection system operates. Additionally, we plan to consider the inclusion of real-world data to improve our study’s realism and practical relevance.

## 5. Conclusions

In this study, we analyzed five semi-supervised learning techniques (IF, LOF, OCSVM, MLP-AE, and Conv-AE) for anomaly (e.g., complete, concurrent, and complex) detection in WWTPs. The comparison between the two autoencoders and the classical semi-supervised methods reveals that AEs offer a significantly enhanced efficiency in addressing anomaly detection challenges within WWTPs. Remarkably, Conv-AE achieved over 98% accuracy, precision, recall, and F1-score in the case of complete and complex anomalies. And in the case of concurrent anomalies, it obtained over 97% accuracy, precision, recall, and F1-score.

Also, through the development of an advanced anomaly detection system based on the optimal semi-supervised method (Conv-AE), our approach not only facilitates the early detection of various anomalies in DO sensor and aeration valve behaviors, but also provides valuable insights to the operators about the system operational status. For example, in the case of complete anomalies, we achieved time delays that are significantly smaller compared with other studies. Moreover, the uniqueness of our study lies in the capability of our anomaly detection system to identify not only complete faults, as commonly presented in the literature, but also concurrent and incipient faults. To our knowledge, this specific aspect has not been studied in other publications so we cannot compare our results with that of other research.

Hence, it becomes evident that embedding semi-supervised learning techniques in anomaly detection holds the power to revolutionize how wastewater treatment plants operate. This means more efficient operations, the better use of resources, and a stronger commitment to protect the environment.

In our future research, we plan to extend our current study to include the analysis of biological faults, further enhancing the AE fault detection system by exposing it to a diverse set of fault types using real-world data. Additionally, we propose to investigate alternative sensor types and explore multi-sensor fault scenarios. Given the complexity of the wastewater treatment process and the numerous sensors involved, incorporating real-world data into our experiments will provide a more comprehensive understanding of the challenges and opportunities in this domain.

## Figures and Tables

**Figure 1 sensors-23-08022-f001:**
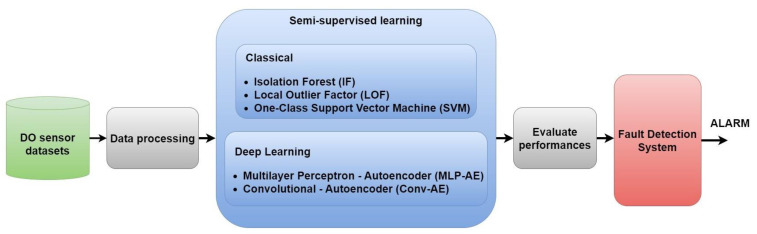
Block diagram of the proposed framework.

**Figure 2 sensors-23-08022-f002:**
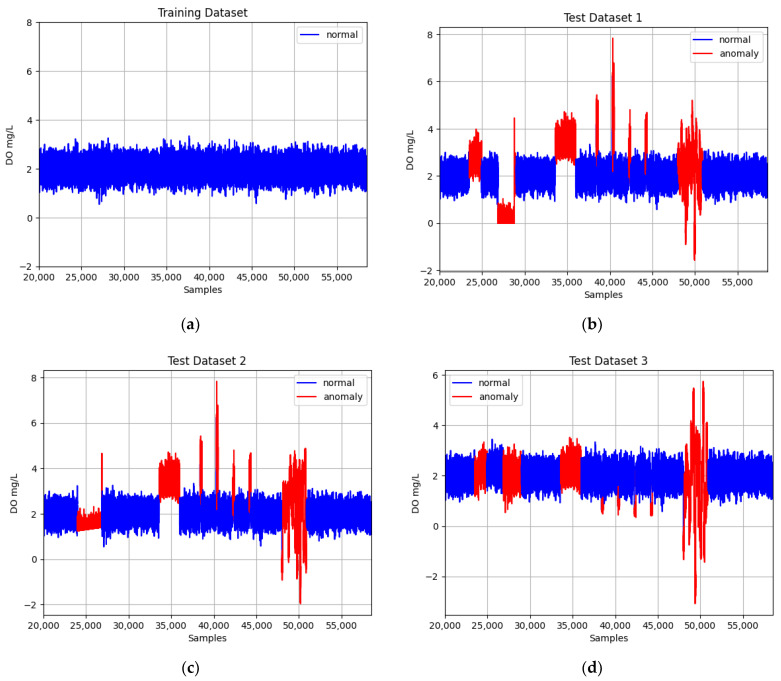
The normal and anomaly data from datasets: (**a**) Training dataset with normal data; (**b**) Test dataset 1 with normal and complete anomalies; (**c**) Test dataset 2 with normal and concurrent anomalies; (**d**) Test dataset 3 with normal and complex anomalies.

**Figure 3 sensors-23-08022-f003:**
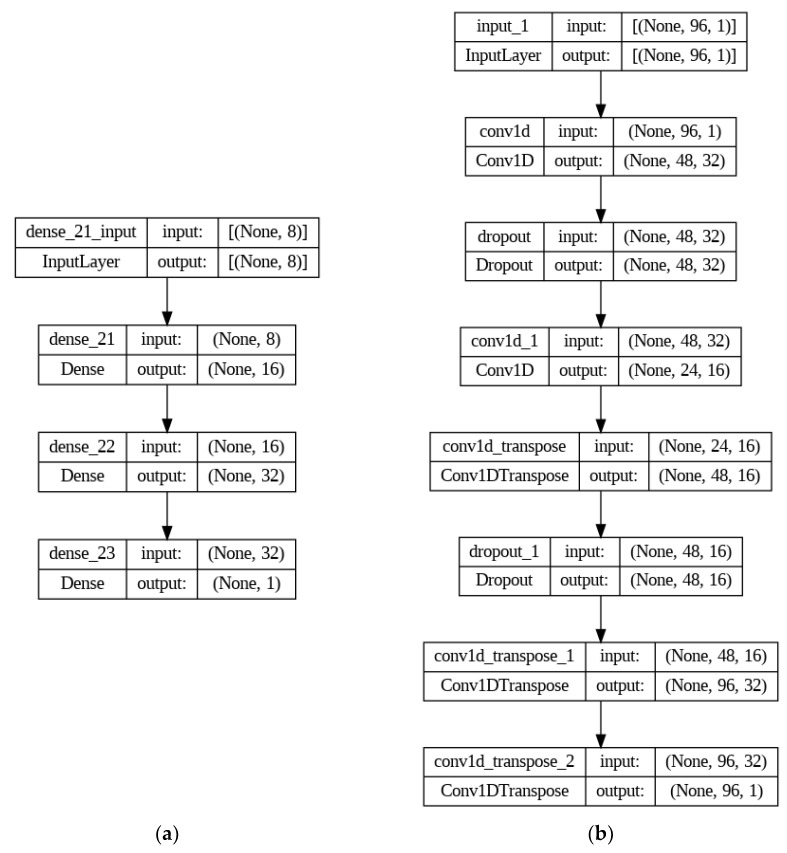
Autoencoder architectures: (**a**) MLP-AE; (**b**) Conv-AE.

**Figure 4 sensors-23-08022-f004:**
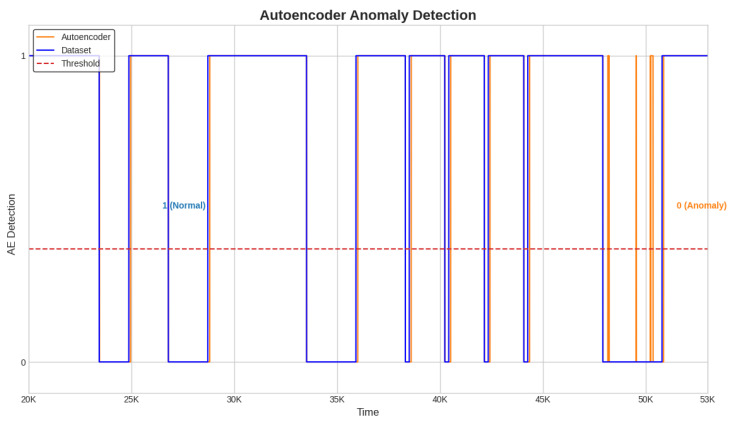
Detection system. Anomaly detection with Conv-AE algorithm; test dataset 1.

**Figure 5 sensors-23-08022-f005:**
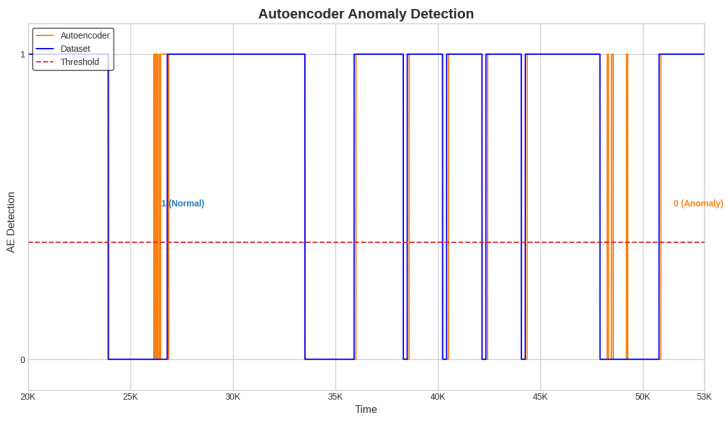
Detection system. Anomaly detection with Conv-AE algorithm; test dataset 2.

**Figure 6 sensors-23-08022-f006:**
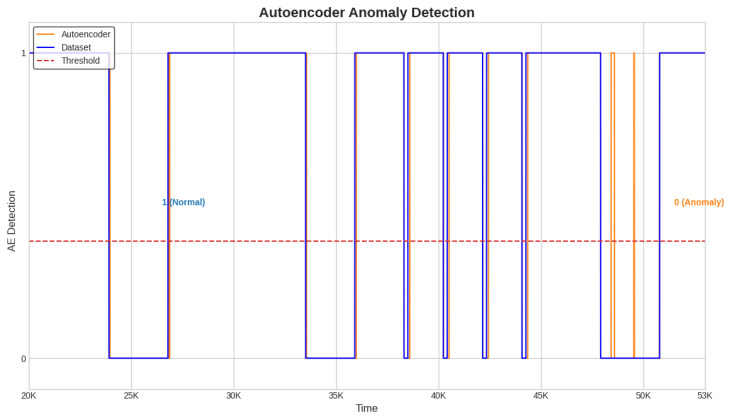
Detection system. Anomaly detection with Conv-AE algorithm; test dataset 3.

**Table 1 sensors-23-08022-t001:** Anomaly scenarios in WWTPs.

Dataset	Anomaly	Start (Day)	Duration (Hours)
Test Dataset 1	Aeration valve	300	480
	Sensor drift	250	720
	Sensor bias	350	600
	4 × Sensor spike	400, 420, 440, 460	48, 48, 48, 48
	Sensor PD	500	720
Test Dataset 2 (Concurrent faults)	Aeration valve + sensor drift	250	720
	Aeration valve + sensor bias	350	600
	4 × (Aeration valve + sensor spike)	400, 420, 440, 460	48, 48, 48, 48
	Aeration valve + sensor PD	500	720
Test Dataset 3 (Concurrent + incipient faults)	80% Aeration valve + 20% sensor drift	250	720
	80% Aeration valve + 20% sensor bias	350	600
	4 × (80% Aeration valve + 20% sensor spike)	400, 420, 440, 460	48, 48, 48, 48
	80% Aeration valve + 20% sensor PD	500	720

**Table 2 sensors-23-08022-t002:** Statistical analysis of the training dataset.

	Time	DO
count	58,465.000000	58,465.000000
mean	304.500180	2.000253
std	175.807709	0.308152
min	0.000000	0.244860
25%	152.250000	1.797200
50%	304.500000	2.002700
75%	456.750000	2.203600
max	609.000000	3.337000

**Table 3 sensors-23-08022-t003:** Hyperparameters of classical SSLs. In cases where hyperparameters are not explicitly specified, they are assumed to take on their default values.

Classical SSLs	Hyperparameters
IF	n_estimators = 100, random_state = 5
LOF	n_neighbors = 20, metric = ‘Euclidean’
OCSVM	Kernel = ‘rbf’, degree = 3, gamma = 0.1, nu = 0.05

**Table 4 sensors-23-08022-t004:** Hyperparameters of AEs.

AEs	Hyperparameters
MLP-AE	Encoder: Three layers: 1st—32 neurons, 2nd—16 neurons, 3rd—8 neurons. All with ReLU activation. Decoder: Three layers: 1st—16 neurons with ReLU activation, 2nd—32 neurons with ReLU activation, and 3rd—1 neuron with Sigmoid activation. Compiled with Adam optimizer, learning rate = 0.001, and MAE loss function
Conv-AE	Encoder: First Transpose Conv. Layer with 32 filters, 7 kernel size, ‘same’ padding, 2 strides, and ReLu activation. Dropout layer: 0.2 rate. Second Transpose Conv. Layer with 16 filters, 7 kernel size, ‘same’ padding, 2 strides, and ReLU activation. Decoder: First Transpose Conv. Layer with 16 filters, 7 kernel size, ‘same’ padding, 2 strides, and ReLU activation. Dropout layer: 0.2 rate. Second Transpose Conv. Layer with 32 filters, 7 kernel size, ‘same’ padding, and ReLU activation. Third Transpose Conv. Layer with 1 filter, 7 kernel size, ‘same’ padding. Compiled with Adam optimizer, learning rate = 0.001, and MSE loss function.

**Table 5 sensors-23-08022-t005:** Evaluation metrics of five SSLs.

Anomaly Scenarios	Semi-Supervised Algorithm	Accuracy	Precision	Recall	F1-Score
Test Dataset 1	IF	79.18	80.68	79.18	79.87
	LOF	73.09	74.97	73.09	73.98
	OCSVM	81.62	75.12	81.62	77.61
	MLP-AE	94.36	94.46	94.36	93.95
	Conv-AE	98.36	98.42	98.36	98.38
Test Dataset 2	IF	79.36	80.75	79.36	80.01
	LOF	75.36	76.97	75.36	76.14
	OCSVM	83.07	77.86	83.07	79.78
	MLP-AE	92.40	92.57	92.40	91.48
	Conv-AE	97.81	97.80	97.81	97.80
Test Dataset 3	IF	78.87	80.28	78.87	79.53
	LOF	75.29	76.91	75.29	76.07
	OCSVM	82.43	76.24	82.43	78.75
	MLP-AE	88.70	88.93	88.70	86.11
	Conv-AE	98.64	98.66	98.64	98.65

**Table 6 sensors-23-08022-t006:** Confusion matrices of five SSLs.

Anomaly Scenarios	IF	LOF	OCSVM	MLP-AE	Conv-AE
Test Dataset 1	[[8521 1380] [1055 737]]	[[8165 1736] [1411 381]]	[[9418 483] [1666 126]]	[[48,795 254] [3044 6372]]	[[48,244 710] [246 9170]]
Test Dataset 2	[[8675 1346] [1067 605]]	[[8441 1580] [1301 371]]	[[9539 482] [1498 174]]	[[49,267 263] [4181 4754]]	[[48,851 584] [694 8241]]
Test Dataset 3	[[8646 1375] [1096 576]]	[[8437 1584] [1305 367]]	[[9529 492] [1562 110]]	[[49,268 262] [6346 2589]]	[[48,933 502] [292 8643]]

**Table 7 sensors-23-08022-t007:** Detection system: time delay of each anomaly is detected with Conv-AE detection system.

Dataset	Anomaly	Time (Hours)
Test Dataset 1	Aeration valve	0.72
	Sensor drift	3.84
	Sensor bias	0.72
	4 × Sensor spike	0.96, 0.48, 1.68, 1.68
	Sensor PD	2.4
Test Dataset 2	Aeration valve + sensor drift	6
	Aeration valve + sensor bias	0.72
	4 × (Aeration valve + sensor spike)	0.96, 0.48, 1.68, 1.68
	Aeration valve + sensor PD	0.48
Test Dataset 3	80% Aeration valve + 20% sensor drift	11.52
	80% Aeration valve + 20% sensor bias	5.52
	4 × (80% Aeration valve + 20% sensor spike)	0.96, 1.68, 1.2, 0.96
	80% Aeration valve + 20% sensor PD	0.48

## Data Availability

The datasets and the Python source code from this study are available here: https://gitfront.io/r/mmigyt/feE6E5AoBe2C/Unsupervised-ML/.

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
