# Peer review of "Semi-Supervised Anomaly Detection of Dissolved Oxygen Sensor in Wastewater Treatment Plants"

_sensors, 2023, doi:10.3390/s23198022_

Round 1

Reviewer 1 Report

It can be accepted after some minor revisions:

1. There are quite few spelling mistakes and sometimes sentences do not make sense. Please review the paper and improve English. Authors whose native language is not English are advised to have their papers reviewed by a colleague who is proficient in English before they are submitted.

2. Some figures is not clear, please redraw them.

3. Critical commentary is needed from the author who eventually recommends directions for further research.

4.This work has cited lots of valuable publications about contact angle, however, some recent related references should be cited as:

(1) https://doi.org/10.1002/jctb.6288

(2) https://doi.org/10.1016/j.cep.2019.107626

(3) https://doi.org/10.1007/s40430-021-02971-0

(4) https://doi.org/10.1016/j.cherd.2019.03.002

It can be accepted after some minor revisions:

1. There are quite few spelling mistakes and sometimes sentences do not make sense. Please review the paper and improve English. Authors whose native language is not English are advised to have their papers reviewed by a colleague who is proficient in English before they are submitted.

2. Some figures is not clear, please redraw them.

3. Critical commentary is needed from the author who eventually recommends directions for further research.

4.This work has cited lots of valuable publications about contact angle, however, some recent related references should be cited as:

(1) https://doi.org/10.1002/jctb.6288

(2) https://doi.org/10.1016/j.cep.2019.107626

(3) https://doi.org/10.1007/s40430-021-02971-0

(4) https://doi.org/10.1016/j.cherd.2019.03.002

Author Response

Thank you very much for taking the time to review this manuscript. Please find the detailed responses in the attached file and the corresponding corrections highlighted in red.

We would like to thank the esteemed Reviewer for reviewing our work. The received comments were extremely useful - they have helped us to improve the quality and the content of our paper. We have tried to rectify all the issues raised by the esteemed Reviewer. Please find attached the point-by-point response to the received comments and suggestions.  

Thank you!

Reviewer 2 Report

This manuscript investigated different machine/deep learning approaches for anomaly detection of dissolved oxygen sensor in wastewater treatment plants, where isolation forest, local outlier factor, one-class support vector machine, multilayer perceptron autoencoder and convolutional autoencoder were employed for the task of interest. The experimental data were used for validation of different learning methods. The results show that Convolutional Autoencoder is capable of achieving the best performance among five models in terms of anomaly detection. Overall, the topic of this research is interesting, and the manuscript was well structured and written. Before it is accepted for publication, I suggest a minor revision by addressing the following comments.

1.       The main novelty and contributions of this research should be well demonstrated in abstract and introduction.

2.       Please broaden and update literature review on machine/deep learing methods (such as autoencoder or convolutional networks) in engineering applications like fault/damage detection. E.g. Automated damage diagnosis of concrete jack arch beam using optimized deep stacked autoencoders and multi-sensor fusion.

3.       It is well known that the performance of machine/deep learning models are related to setting of hyperparameters. How did the authors assign model parameters in this research to conduct a fair comparison?

4.       In this paper, the authors only provided the architecture of multilayer AE and convolutional AE. Please give the architecture of other machine learning models as well in terms of diagram.

5.       It is better to use confusion matrix to demonstrate the model performance.

6.       The running time should be considered as well as one of evaluation metrics for practical application.

7.       More future research should be included in conclusion part.

Author Response

Thank you very much for taking the time to review this manuscript. Please find in the attached file the responses and the corresponding corrections highlighted in red.

We would like to thank the esteemed Reviewer for reviewing our work. The received comments were extremely useful - they have helped us to improve the quality and the content of our paper. We have tried to rectify all the issues raised by the esteemed Reviewer. Please find attached the point-by-point response to the received comments and suggestions.  

Thank you very much!

Reviewer 3 Report

Dear Authors:

In this study, the authors evaluated five SEMI-supervised machine-learning techniques (Isolation Forest, Local Outlier Factor, One-Class Support Vector Machine, Multilayer Perceptron Autoencoder, and Convolutional Autoencoder) to detect various anomalies (complete, concurrent, and complex) in the Dissolved Oxygen (DO) sensor and aeration valve at a WWTP. The Convolutional Autoencoder algorithm yielded the best results, achieving an accuracy of 98.36% for complete faults, 97.81% for concurrent faults, and 98.64% for complex faults (combining incipient and concurrent faults). However, there are several major issues to be addressed in this paper.

1) Title: The applied approaches semi-supervised, not unsupervised. In the paper, it is crucial to accurately depict the nature of the applied approaches as semi-supervised, rather than unsupervised. These methods necessitate nominal data during training, which aligns them with semi-supervised techniques. It is recommended to explicitly define and clarify this aspect in the paper to provide a more accurate representation of the methodology employed.

2) In the abstract, it is important to provide a clear and concise overview of the core work and innovations of the study. Including step-by-step details of the method can help readers better understand the unique contributions and approach of the research. 

3) In introduction part, you have to reconstruct it according to the sequence of background, problem to state, existing methods with disadvantages, your method and from which point to solve. I find

the related works are disorder.

4) The related work section should be improved by discussing other similar studies on anomaly detection in WWTP; Here are some recent studies and you can find other: 1) DOI: 10.1109/ACCESS.2019.2933616 ; 2) DOI: 10.1109/DDCLS55054.2022.9858416

5) The presented methods should be clearly described

6) statistical analysis of training data without anomalies need to be included after the data description. 

6) The evaluation should be done with anomaly with different magnitude levels to see the sensitivity of these approaches

7) The discussion should be improved and the complexity of these approaches should be reported 

8) Limitations of these approaches need be discussed

The language of the paper can be checked to avoid typos.

Author Response

(The authors gave the same response as above.)

Round 2

Reviewer 1 Report

accept

Author Response

Thank you very much for the positive response to our paper!

We appreciate very much your help in improving our study during the review report from round 1.

Thanks and Regards,

Mihaela Miron

Reviewer 3 Report

Dear Authors,

There is a clear confusion in the paper. In addition, if you are using ChatGPT to obtain a definition, this is not the appropriate method. There are numerous scientific references available that provide clear definitions of semi-supervised methods. See, for example, Qi, R., Rasband, C., Zheng, J. and Longoria, R., 2021. Detecting cyber attacks in smart grids using semi-supervised anomaly detection and deep representation learning. Information, 12(8), p.328.

Here is the definition of a semi-supervised approach:  'Semi-supervised Anomaly Detection also utilizes both training and test datasets. However, the training data exclusively comprises normal data without any anomalies. The fundamental concept behind this approach is to learn a model of the normal class, allowing anomalies to be detected when they deviate from that model. This concept is also referred to as "one-class" classification.'  You can find a more detailed distinction between supervised, semi-supervised, and unsupervised methods in the following paper, page 10:

Chandola, V., Banerjee, A. and Kumar, V., 2009. Anomaly detection: A survey. ACM computing surveys (CSUR), 41(3), pp.1-58.

The correction should be applied throughout the entire paper, starting from the title.

Regards,

Minor editing of English language required.

Author Response

Thank you very much for the second review of this manuscript!

The received comments from round 2 were very helpful as they allowed us to fine tune further our paper. We have modified accordingly all the issues raised by the esteemed Reviewer.

Please find attached the file with the point-by-point response to the received comments and suggestions.  

Thanks and Regards,

Mihaela Miron
